# Incidence and Death Rates from COVID-19 Are Not Always Coupled: An Analysis of Temporal Data on Local, Federal, and National Levels

**DOI:** 10.3390/healthcare9030338

**Published:** 2021-03-17

**Authors:** Stefan Borgmann, David Meintrup, Kerstin Reimer, Helmut Schels, Martina Nowak-Machen

**Affiliations:** 1Department of Infectious Diseases and Infection Control, Ingolstadt Hospital, 85049 Ingolstadt, Germany; stefan.borgmann@klinikum-ingolstadt.de; 2Faculty of Engineering and Management, Technische Hochschule Ingolstadt, 85049 Ingolstadt, Germany; David.Meintrup@thi.de; 3Statistical Office of the City of Ingolstadt, 85049 Ingolstadt, Germany; kerstin.reimer@ingolstadt.de (K.R.); Helmut.schels@ingolstadt.de (H.S.); 4Department of Anesthesia and Intensive Care Medicine, Ingolstadt Hospital, 85049 Ingolstadt, Germany; 5Teaching Faculty of the Department of Anesthesiology at University Hospital of Tuebingen, 72072 Tuebingen, Germany

**Keywords:** time series analysis, COVID-19 incidence, case-fatality rate, herd immunity, Upper Bavaria

## Abstract

SARS-CoV-2 has caused a deadly pandemic worldwide, placing a burden on local health care systems and economies. Infection rates with SARS-CoV-2 and the related mortality of COVID-19 are not equal among countries or even neighboring regions. Based on data from official German health authorities since the beginning of the pandemic, we developed a case-fatality prediction model that correctly predicts COVID-19-related death rates based on local geographical developments of infection rates in Germany, Bavaria, and a local community district city within Upper Bavaria. Our data point towards the proposal that local individual infection thresholds, when reached, could lead to increasing mortality. Restrictive measures to minimize the spread of the virus could be applied locally based on the risk of reaching the individual threshold. Being able to predict the necessity for increasing hospitalization of COVID-19 patients could help local health care authorities to prepare for increasing patient numbers.

## 1. Introduction

According to the World Health Organization (WHO), the pandemic pathogen SARS-CoV-2 has caused more than 1,843,000 deaths worldwide [1]. Infection rates with SARS-CoV-2 and the related mortality of COVID-19 are not equal among countries. Apart from varying death numbers on a national level, significant regional differences have occurred in neighboring regions [2].

In spring 2020, governments of many countries, predominately in the Northern Hemisphere, locked down public life to avoid the spread of the virus. Although these measures proved to be effective in lowering morbidity and mortality [3], lockdown measures resulted in many negative effects such as negative economic impact, reduced physical activity, increased obesity, and tobacco and alcohol consumption [4,5,6]. Furthermore, many people suffered from social restrictions leading to psychological turmoil and emotional distress [5].

Many European countries have currently imposed a second lockdown following a massive surge of SARS-CoV-2 infections over the late fall and early winter months in 2020.

In many parts of the world, social restriction measures, as well as closure of restaurants and shops to minimize infections, are currently regulated based on SARS-CoV-2 infection numbers during a 7-day period per 100,000 people [7]. However, the mortality from COVID-19 and/or the case-fatality rate (CFR) is rarely considered. Bertuzzo et al. recently used the number of hospitalizations from COVID-19 to analyze the geographical spread of SARS-CoV-2 [3].

The spread of the virus is dynamic and changes from country to country as well as from community to community on a small scale. A phenomenon that can currently be observed all over the world is that countries, in an effort to find common political strategies, create lockdown measures that may be too harsh for certain regions based on their infection rates, and too mild for regions that are more severely impacted by COVID-19 infections. The actual number of COVID-19 infections with a resulting case-fatality rate in a certain geographical region may not necessarily match the case-fatality rate in a neighboring district or even neighboring town. In order to develop strategies to minimize general public and economical restrictions but to guarantee maximal safety of the population as well as functionality of local health care systems, it seems necessary to analyze infection rates and resulting hospitalizations as well as case-fatality rates on a local level as opposed to a national average.

In the present study, we performed a timeline analysis since the beginning of the pandemic until December 2020 to study the relation between incidence of SARS-CoV-2 infections and the actual case-fatality rate (CFR) of COVID-19. The analysis was performed on the population of three administrative levels, ranging from country (Germany) through state (Bavaria) to a local city (Ingolstadt), to further explore the question of whether COVID-19 infections can be anticipated on a local level, and whether local restrictive measures can be announced accordingly in order to protect the functionality of local health care systems and populations.

## 2. Methods

### 2.1. Data Acquisition

Bavaria is Germany’s largest federal state by surface area, with 13 million inhabitants, located in southeastern Germany bordering Austria and close to northern Italy (Figure 1). Throughout the course of the pandemic Bavaria has constantly been the state with one of the highest infection rates in Germany.

Upper Bavaria exhibits the highest population density (267 inhabitants/km^2^) in Bavaria. The “Area 10” is a district located just north of Munich in Upper Bavaria and is composed of the city of Ingolstadt (137,000 inhabitants), the rural regions of Eichstätt (133,000 inhabitants), Neuburg-Schrobenhausen (97,000 inhabitants), and Pfaffenhofen (128,000 inhabitants). Ingolstadt hospital is the major public city hospital of the region with 1077 hospital beds and about 50 intensive care unit (ICU) beds serving about 500,000 people.

In Germany, COVID-19 is a disease that mandates reporting to the local health authorities. Total cases of COVID-19 patients in Ingolstadt were obtained from the statistical office of Ingolstadt city. The number of COVID-19 patients treated in Ingolstadt hospital and the number of COVID-19 patients treated in the ICUs were obtained from hospital recordings. The number of COVID-19 patients treated in the ICUs for the whole of Germany was provided by the German Interdisciplinary Association for Intensive Care and Emergency Medicine (DIVI) [8].

The incidence of COVID-19, the incidence of COVID-19-caused deaths, and the case-fatality rate (CFR) in Germany as well as in Bavaria per 100,000 inhabitants was calculated using published data from the Robert Koch Institute, the European Centre for Disease Control, the Federal Statistical Office of the Federal Republic of Germany, and the Bavarian Health and Food Safety Authority [7,8,9,10,11].

### 2.2. Statistical Analysis

The temporal relation of COVID-19 incidence $\left(x_{t}\right)$ and COVID-19-caused deaths ($y_{t}$) was analyzed using the following intuitive time series model: (1)$$y_{t}=\alpha \cdot x_{t-d}$$

In this model, $\left(x_{t}\right)$ is the 7-day moving average of COVID-19 incidences, and $\left(y_{t}\right)$ is the 7-day moving average of COVID-19-related deaths on day t. The model parameters α and d correspond to the estimated percentage of incidences leading to death and the shift in days between reported infection and death, respectively. The shift parameter d takes care of the right censoring of the data, and allows one to estimate proper case-fatality rates α. The model was fitted to the data using a minimum error variance approach.

Figure 2 shows an example of this calculation for the first 6 months of the pandemic in Germany. The model (black line in Figure 2) is given by:(2)$${\hat{y}}_{t}=0.045\cdot x_{t-14}$$

The model fits the actual deaths (red line in Figure 2) very well until the end of June. From this point on, the model prediction (black dotted line in Figure 2) and the actual deaths uncouple. Despite increasing incidences in July and August of 2020, we did not observe an increase in COVID-19-related deaths, as the model predicts. Therefore, we changed the model for the summer months, assuming a constant death rate independent of the incidences. Later, during the second wave, incidences and deaths are again coupled, but with a different case-fatality rate. Therefore, our final model takes the following form:$$y_{t}=\left\{\begin{array}{cc} \alpha_{1}\cdot x_{t-d_{1}}\; & \mathrm{during}\;\mathrm{the}\;\mathrm{first}\;\mathrm{wave},\\ p_{2}\; & \mathrm{during}\;\mathrm{the}\;\mathrm{summer},\\ \alpha_{2}\cdot x_{t-d_{2}}\; & \mathrm{during}\;\mathrm{the}\;\mathrm{second}\;\mathrm{wave}. \end{array}\right.$$

It is important to note that we did not prescribe the time intervals for these three phases. The beginning and end of the second phase with a constant, unconnected death rate was part of the optimization of the model and chosen in a way to minimize the error variance. 

## 3. Results

On 21 December 2020, the German Robert Koch Institute reported a cumulative of 1,530,180 cases in Germany resulting in 27,006 deaths (CFR 1.76%) [10]. Health authorities of Bavaria communicated a cumulative of 292,899 cases resulting in 5821 deaths (CFR of 1.99%), matching national data [9]. Upper Bavaria, where Ingolstadt is located, reported the highest number of infections (114,750) and deaths (1928) within Bavaria. However, the highest CFRs occurred in Upper Palatinate (2.51%) and Upper Franconia (2.48%). Interestingly, when comparing the seven Bavarian government districts, population density was negatively correlated with COVID-19-associated CFR. (Figure 1).

A time series analysis showed the temporal incidence of SARS-CoV-2 infections and the incidence of COVID-19-related deaths in Germany (Figure 2). A prediction model for COVID-19-associated deaths based on actual incidences was estimated for Germany. The model fit is excellent for the first part of the pandemic in the spring of 2020, where the actual reported deaths matched the predicted deaths and followed the incidence with a predicted timely delay (Figure 2), parallel incidence followed by black and red lines, showing a coupling of all curves. After June 2020, the incidence of new infections started to rise and re-peaked in the late summer. Whereas the model predicted a similar increase for expected deaths based on data from the spring (black dotted line), the actual mortality curve surprisingly remained flat until the end of October of 2020. This phenomenon can be described as an uncoupling of the previously coupled incidence rates and the resulting deaths.

Taking into consideration the experience from the summer where infection rates did not match the expected death rates in Germany (uncoupling), we changed the statistical model to a 3-phase model, with coupled death rates during the first and second wave and a constant death rate in between.

For the second wave of the pandemic, starting in October of 2020, we now applied this adapted model to the level of a country (Germany), a federal state (Bavaria), and a district city (Ingolstadt). A summary of the estimated model parameters is given in Table 1; the corresponding graphs are depicted in Figure 3.

We found that our model correctly predicted the rising of the death rate in all three cases in the first and second wave. Interestingly, we found an uncoupling of the infection rates and the anticipated death rates in all geographic regions over the summer with generally low infection rates, which means that rising infection numbers did not result in the expected rise of deaths from COVID-19. The model correctly predicted the recoupling of infection rates and death rates in all three geographic regions in the fall at the beginning of the so-called second wave. However, all three geographic regions showed different time points for recoupling. First, we found a recoupling on a national level (Germany mid-October), followed by the state (Bavaria end-October), and at last by the district city (Ingolstadt early November), which for a while lagged behind the national average of deaths despite rising infection numbers. In each geographic model an individual threshold of infection rates seemed to be necessary in order to trigger the deaths to rise and the curves to recouple. By anticipating the recoupling point as done in our model, rising death rates and hospitalizations might be predicted on a local level. Local health care systems and authorities could then prepare and launch specific geographically adapted measures of social restrictions (Figure 2).

Intensive care unit occupancy data were retrieved from DIVI [8] and from the documentations system of Ingolstadt hospital. The data show that ICU admissions during the first wave were much higher when set in relation to the net number of total infections. However, ICU admissions are rising at this point in time and might be following a threshold similar to the general death rate.

Infection rates (grey curves, left scale) of COVID-19 for Germany, Bavaria, and Ingolstadt were recorded from the first wave of the pandemic in March 2020 over the summer where low infection rates were reported until the beginning of the second wave with surging infection rates end of October/beginning of November 2020. The temporal course of infections matches in all three geographic regions (Figure 3).

Our prediction model depicts the official actual death rates (red curve, right scale) and the calculated anticipated death rates per geographic region (black curves, right scale). Coupled incidence and actual death curves (black and red curves) are found in all geographic models after a certain individual delay. The calculated anticipated death curve is correctly coupled over time. An uncoupling of the incidence and actual death curves can be observed from July until October. During this time, slightly rising infection numbers did not result in rising deaths. Once a certain individual threshold is reached, all three curves are recoupled following an individual timeline.

ICU admissions are shown (blue line, left scale) and show a similar pattern as the infection and death rate curves. ICU admission numbers seem to be rising slower in the second wave but might follow a similar threshold pattern with rapidly increasing numbers once the threshold is reached.

Comparing case-fatality rates (CFR) of adjacent regions, it becomes obvious how differently the course of the pandemic evolves in neighboring regions. The city of Ingolstadt and the surrounding districts are only 10–20 km apart. Whereas both regions were roughly equally affected during the first phase of the pandemic, the second wave hit the region much harder than the city of Ingolstadt (Figure 4). Applying the CFR of the region during the second wave to the city of Ingolstadt (pink curve in Figure 4) would have resulted in significantly more deaths than the actual reported ones. The importance of individual regionally adaptable models for the prediction of CFRs seems obvious.

In order to examine the impact of COVID-19-caused mortality on total mortality of our population, the number of individuals deceased in 2020 was compared to the average number of deceased individuals between 2016 and 2019. In general, mortality in 2020 was higher than in earlier years on a national, federal, and local level. Until week 47, excess mortality in Germany was 12,903, of which 12,016 deaths were attributed to COVID-19. A similar result was observed for Bavaria and Ingolstadt, where excess mortality was 4703 (Bavaria) and 86 (Ingolstadt) and SARS-CoV-19 was accused for 4303 (Bavaria) and 45 (Ingolstadt) deaths, respectively. The chronological sequence of excess mortality and SARS-CoV-2-attributed deaths is shown in Figure 5. The number of deaths was substantially lower in the first weeks of 2020 when compared to previous years. However, after SARS-CoV-2 started to spread within our population, death numbers increased and were ultimately higher than in earlier years. While the number of COVID-19-caused deaths and excess mortality followed a similar course in the spring (first wave) and fall (second wave), the excess mortality remained high over the summer months when only low numbers of COVID-19-related deaths were reported.

Staff infections with COVID-19 in health care facilities have been widely reported all over the world. Especially, hospital staff taking care of COVID-19 patients are particularly at risk. Different protective measures have been implemented in order to protect staff from getting infected. N-95 masks seem to be one of the most effective measures to protect staff from aerosolized virus transmission [12]. 

## 4. Discussion

Our data show that COVID-19-related case-fatality rates have not been homogenous in Germany when examining the first and second waves of the COVID-19 pandemic and its spatial and temporal properties. In addition, we found that despite high numbers of COVID-19 infected individuals at the beginning of the second wave, overall case-fatality rates remained low when compared with the first wave. Following this finding, we developed a statistical “case-fatality prediction model”, with the idea of developing a tool to help officials in the decision-making process for the implementation of lockdown measures to slow the spread of the virus. The key to understanding our model is that local case-fatality rates in individual towns or small geographical areas are being predicted, as opposed to predictions based on national or federal calculations. 

We were able to show that in every geographical area that we examined, a certain individual threshold of infected individuals needed to be surpassed in order to trigger increasing case-fatality rates. Those infection thresholds differed from area to area and could not be interchanged.

One explanation for these significant differences in CFR between areas is certainly based on local demographics. An area with a predominantly elderly population will most likely have a lower threshold than an area mainly composed of young healthy families [13,14]. However, the minority of towns are composed of a homogenous demographic population. As a consequence, knowing these individual thresholds might provide one part of the puzzle when planning for hospital capacities and restrictive measures. We need to expand our studies further in order to describe and quantify this individual threshold in more detail. However, the concept of localized individual infection thresholds that trigger rising case-fatality rates is novel and in our opinion deserves attention. 

The infection threshold in a local community is not only important to know because of its implications for the following expected increase in deaths but also because of marks in shift in the aggressiveness of the disease. Before the increase of the actual case-fatality rates we always observe an increase in hospital admissions as well as ICU occupancy [8]. The goal must therefore be to detect the critical individual infection threshold and then immediately apply localized restrictive measures such as temporary lockdowns, school closings etc. to actually prevent case-fatality rates from rising and hospital capacities from being limited.

We are aware that more detailed analyses composed of a broader geographical area need to be conducted to validate our case-fatality prediction model. However, assuming that our model correctly predicts case-fatalities based on local circumstances, strict contact limitations during the summer months, especially during the months with low infection rates, were probably not necessary and did not prevent any fatalities seen in the following second wave. Our data suggest that a certain individual geographical threshold of SARS-CoV-2 infections needs to be surpassed in order to generate an increase of case-fatality rates. As a consequence, restrictive social measures could be handled locally, allowing for limited social and economic secondary damage in the future. 

Undermining this theory of infection thresholds for the violent spread of infectious agents, a recent analysis of vancomycin-resistant *Enterococcus faecium* dissemination showed that the spread of the bacteria exhibited features similar to an electrical circuit [15]. The electrical circuit is either turned “on” or “off”. Our model of COVID-19 prediction thresholds shows similarities to an electrical circuit; when a certain threshold of infections is being reached, an increase in CFR seems unavoidable unless the spread of the virus is stopped immediately and aggressively. Such a model could help to better understand the features of COVID-19 epidemiology, especially the phenomenon of a threshold-driven CFR. 

Our analysis of regional data shows that the virus attacked Bavaria earlier than the other parts of Germany. Possibly, a local event contributed to the initial import of the virus from China [16]. Another local event in March resulted in a large outbreak affecting hundreds of individuals in northeastern Bavaria [17]. Ingolstadt registered the first infections at the end of March 2020. Infection rates then increased exponentially in April 2020, similar to the German average.

According to the national health authorities at the Robert Koch Institute, the city of Ingolstadt was exhibiting the highest incidence of infections in all Germany on 1 September Dashboard [18]. Despite the high number of infected individuals, there were no COVD-19 patients in the Ingolstadt hospital. This finding also points towards a decreased pathogenicity of the virus in the summer months in 2020.

In Western Europe, the summer of 2020 was defined as the period in-between peaks of SARS-CoV-2 infections. However, in the late summer months the case-fatality rates steadily declined while infection numbers were beginning to re-rise in Western Europe and specifically in Germany. Interestingly, this phenomenon could be observed internationally and was not limited to Germany or Bavaria [19]. For example, in the U.S., beginning in the summer of 2020, healthcare workers also saw unprecedented increases in COVID-19 diagnoses and hospitalizations, but there wasn’t a congruent rise in mortality rates even as case counts set records. In fact, the COVID-19 mortality rate in the U.S. since the start of the pandemic had decreased at this point [20].

The reason for fewer hospital admissions and SARS-CoV-2-related fatalities despite rising infection rates in the summer remains unknown at this point. One explanation might be that unrecognized herd immunity of the population led to less severe courses. However, on 23 September the German Robert Koch Institute reported a cumulative of 275,927 infections and, therefore, an infection rate of 0.33% based on the total German population [12]. It is highly unlikely that this low percentage of individuals having undergone a SARS-CoV-2 infection is sufficient to mediate herd immunity and to explain the low fatality rate. At the end of October 2020, infection rates started to surge again all over Germany. However, hospitalizations and deaths remained well below the expected numbers known from the first wave. 

Recently, it was reported that in the United States SARS-CoV-2 infections have presumably been underestimated in the early phase of the pandemic [21]. Although some authors question the occurrence of reinfections [22] it is likely that a similar underestimation of infections took place in Europe and some of the current cases represent milder reinfections [23]. When comparing the CFR from April and August 2020 in European countries, decreasing death rates in six out of the seven industrial countries over the course of the pandemic can be observed, despite increasing infection rates [24]. That finding confirms our observation that individuals who were affected in the early phase of the pandemic had a higher risk of dying than those infected later but before the second wave of the pandemic. However, the observation of decreasing CFR despite high infection rates in a population challenges the paradigm of infection as a “yes or no phenomenon” with full immunity after infection. Low-grade circulation of the virus within the population during the summer months of 2020 might have caused limited immunity in contacted individuals. As indicated by the high incidence of COVID-19 in the fall months of 2020, low-grade immunity was not sufficient to prevent the second wave of infections. 

Since the early stages of the pandemic, various and novel pharmacological treatments have been studied and made available for physicians treating COVID-19 patients worldwide, which in certain populations might have led to a decrease in mortality over the course of the pandemic and might explain the decreased CFR despite the high infection numbers [25]. In addition, organizational structures have been implemented and improved when compared to the second wave. Hospital overcrowding and lack of medical equipment was much less common in the second wave compared to the first. Germany was one of the first European countries to implement a national system to register all COVID-19 ICU admissions together with individual hospital capacities in real time on a daily basis [8]. Luckily, Germany at no point during this pandemic had reached the point where patients had to be rejected from hospital admission due to limited capacities in hospital beds, nurses, or medical equipment. When analyzing the data from the National ICU bed registry (“DIVI”) [8], the overall ICU occupancy was stable from April 2020 until February 2021, without major peaks nationally. Of course, regional occupancies changed significantly over the course of the pandemic. However, patients could always be successfully transferred to other facilities to avoid hospital overcrowding. 

Interestingly, the overall ICU mortality in Germany despite improved and novel medical treatments has remained stable at around 30% in Germany over the course of the pandemic until the present day [8]. The fact that the ICU mortality has not significantly changed in Germany over the course of the pandemic points to the hypothesis that CFR might be mainly determined before hospital and, more importantly, before ICU admission. Defining the dangerous threshold geographically, as we postulate in our data, might help to limit the spread of the virus in order to limit hospital admissions. 

In addition, the demographics of infected individuals had changed over the summer months. When infection numbers started to climb at the end of the summer in 2020, mainly younger individuals, often families returning from summer vacation abroad, were among the infected who were less likely to encounter a severe course of the disease [9,12]. As the second wave progressed, these demographics changed and the elderly population became increasingly infected with then-surging CFRs. 

One other important change had taken place between the first and second waves: In all German hospitals, per national mandate, surgical facemasks had become mandatory with escalation to mandatory N-95 masks during certain times in certain hospitals. Since the beginning of the pandemic, various escalation levels and variants of personal protective equipment (PPE) have been discussed. In addition, in–hospital transmissions of SARS-CoV-2 between staff members have been reported and discussed widely over the media. Our data show that the number of staff infections in our hospital significantly declined in the second wave of the pandemic. This was the case even though infection rates were higher when compared to the early months of 2020, when protective equipment was not broadly available and its protective properties had not yet been confirmed. The most likely reason for the decreasing number of staff infections lies in the mandatory introduction of N-95 masks in late June of 2020.

Another interesting finding arises from our data when examining excess mortality, addressing the issue of “undertesting” of the population and as a result, an underestimation of COVID-19-related deaths. Interestingly, SARS-CoV-2-related deaths followed a similar course as the excess mortality during the spring and fall peaks of the pandemic, meaning excess mortality was high when COVID-19-related mortality was high. In contrast, excess mortality remained high over the summer months in 2020 when reported deaths from COVID-19 were at a historical low. We hypothesize that the persistence of high excess mortality over the course of the pandemic might have been attributed to collateral damage from lockdowns and isolation measures such as untreated cardiovascular disease or rapid progression of untreated cancer. In northern Italy, only 52% of the excess mortality during the spring epidemic could be explained with COVID-19-caused deaths. Among the elderly population (>85 years) the percentage of non-COVID-19-caused excess mortality was reported to be 63%. Similar to our study, in Italy, excess mortality was negative within the first weeks of the year [26].

Lastly, since the detection of SARS-CoV-2, a multitude of genetic alterations has been published and three data banks were established for the documentation of the analyses of the corresponding sequences [27]. Apart from the regional distribution of clades, the prevalence of certain clades varies over time. A recently published study hypothesized that in temperate European countries SARS-CoV-2 exhibits higher mutation rates, leading to higher pathogenicity [28], which might change our statistical model and individual infection thresholds based on our model in the future. In addition, undiagnosed mutations might have influenced the epidemiological pattern of the virus in the second wave and might be responsible partly for the higher infection rates specifically among young individuals secondary to a higher pathogenicity. However, due to the variety of political, administrative, and health care systems across Europe that may all influence the CFR to different degrees, it seems unlikely that a single association of certain clades with CFR can be established.

On 20 September 2020, the first SARS-CoV-2 of the B.1.1.7 lineage was detected in Great Britain [29]. Viruses belonging to this clade were stated to be up to 70% more transmissible than other viruses [30]. In southern Germany this virus variant was detected on 24 December 2020 for the first time [31]. Therefore, it is not highly probable that this variant influenced epidemiology of SARS-CoV-2 within the observation period. However, as sequencing for virus mutations had not been routinely implemented in national German testing strategies, more transmissible mutations in theory might have contributed to the high infection numbers in the second wave [32].

Our study has limitations. Analyses are based on observational data from one city and one city hospital. Therefore, the outcome might be biased or driven by an unknown confounder. Another limitation is the accessibility of public health data. In order to apply our statistical model broadly, detailed data need to be collected from local health care authorities that might not be widely accessible.

## 5. Conclusions

The spread of SARS-CoV-2 and the resulting case-fatality rate have not been homogenous across continents, countries, or even defined geographical areas during the current pandemic. Based on this finding, we developed a statistical “case-fatality rate prediction model” with the goal to help officials in the decision-making process for social restrictive measures such as the implementation of lockdowns. We did establish a model that, based on local infection rates and related fatalities, points toward a certain threshold of infections that is needed to trigger a local increase of case-fatality rates. If infection rates stay below this individual threshold, COVID-related hospital admissions seem to be less likely. A less strict approach towards social distancing might then be possible. In order to apply our model to communities across countries, extensive access to public health data is necessary, which might not be available in all areas. However, the novel concept of regional infection rate thresholds for predicting case-fatalities seems like a tool worth adding to the complex mosaic for conquering the spread of the virus and a resumption of an acceptable social life.

## Figures and Tables

**Figure 1 healthcare-09-00338-f001:**
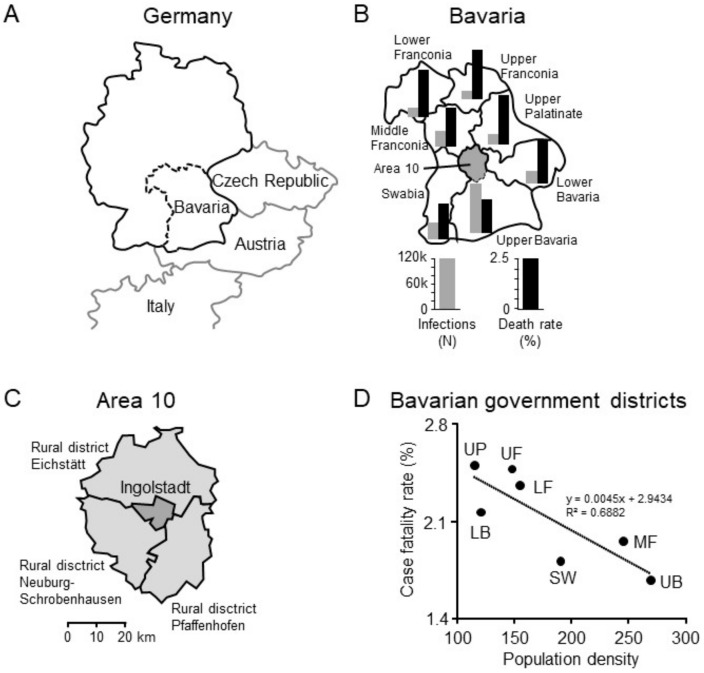
COVID-19 incidence and case-fatality rates in Bavaria, the Bavarian districts, and locally in the city of Ingolstadt. (**A**) Bavaria is Germany’s largest federal state, bordering Austria, the Czech Republic, and lies in close proximity to northern Italy. (**B**,**C**) The geographic “Area 10” of Bavaria, located in the northern part of Upper Bavaria, is composed of the city of Ingolstadt and three rural districts totaling about 500,000 inhabitants. (**D**) Case-fatality rates of Bavarian government districts correlate negatively to the corresponding population densities. LB = Lower Bavaria; LF = Lower Franconia; MF = Middle Franconia; UB = Upper Bavaria; UF = Upper Franconia; UP = Upper Palatinate.

**Figure 2 healthcare-09-00338-f002:**
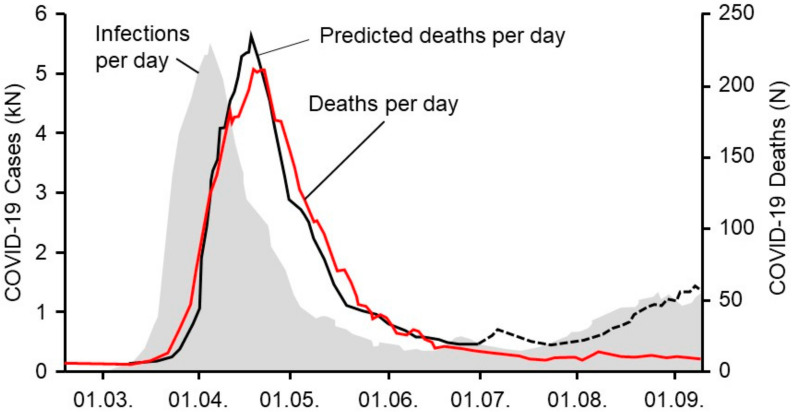
Time series analyses performed on the temporal incidence of SARS-CoV-2 infections (grey area, left scale) and the incidence of COVID-19-associated deaths (red line, right scale). A prediction model (black line, right scale) for COVID-19-associated deaths based on actual incidences was estimated for Germany. The model fit is excellent for the first part of the pandemic. After June 2020, the incidence of new infections started to rise and re-peaked and the model predicted a similar increase for expected deaths based on data from the spring (black dotted line).

**Figure 3 healthcare-09-00338-f003:**
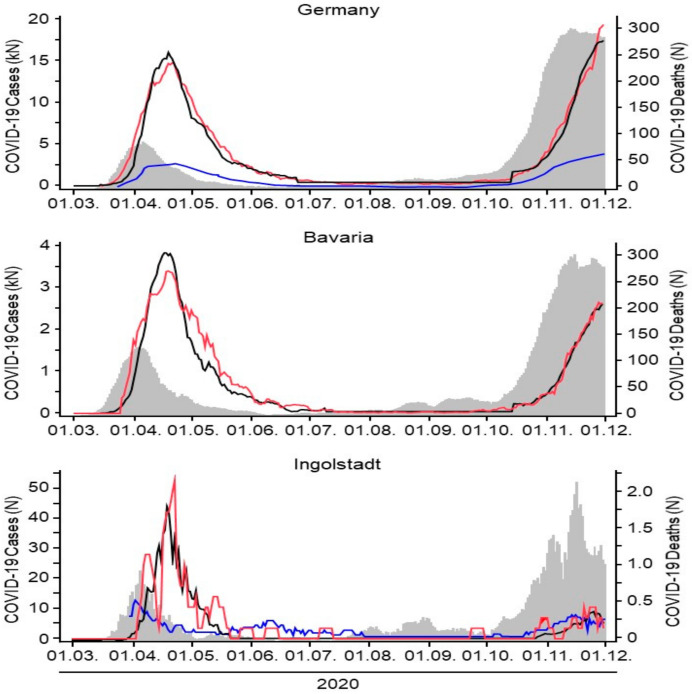
Time series analyses performed on the temporal incidence of SARS-CoV-2 infections (grey area, left scale) and the incidence of COVID-19-associated deaths (red line, right scale). The prediction model (black line, right scale) for COVID-19-associated deaths in Germany, Bavaria, and the city of Ingolstadt. Blue lines: number of patients in Germany and Ingolstadt treated per day on an intensive care unit.

**Figure 4 healthcare-09-00338-f004:**
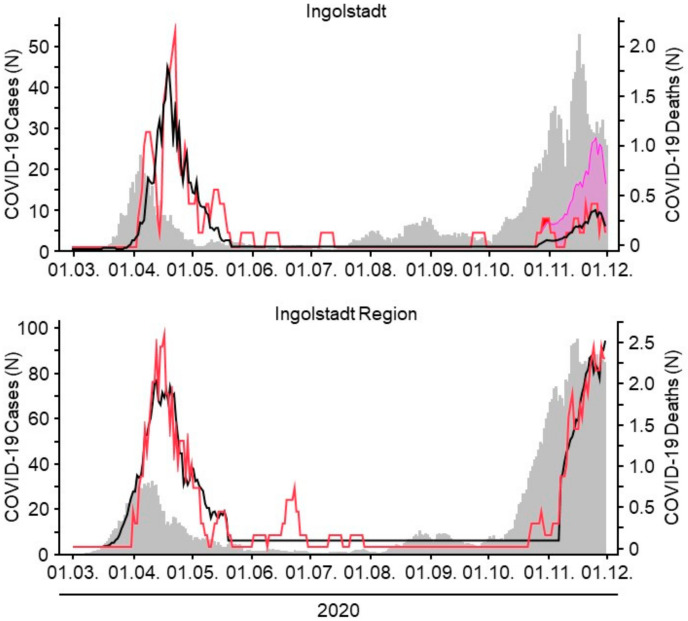
Infection rates (grey curves, left scale) as well as actual case-fatality rates (CFR) (red curve, right scale) and calculated predicted CFRs (black curve, right scale) of COVID-19 for the city of Ingolstadt and the adjacent regional districts (Region) are shown. The case-fatality rate in Ingolstadt during the second wave is significantly lower than in the adjacent Region. The pink curve shows the hypothetical current CFR for Ingolstadt when applying the CFR of the Region, showing that differences in COVID-19 deaths between neighboring regions can be a matter of concern.

**Figure 5 healthcare-09-00338-f005:**
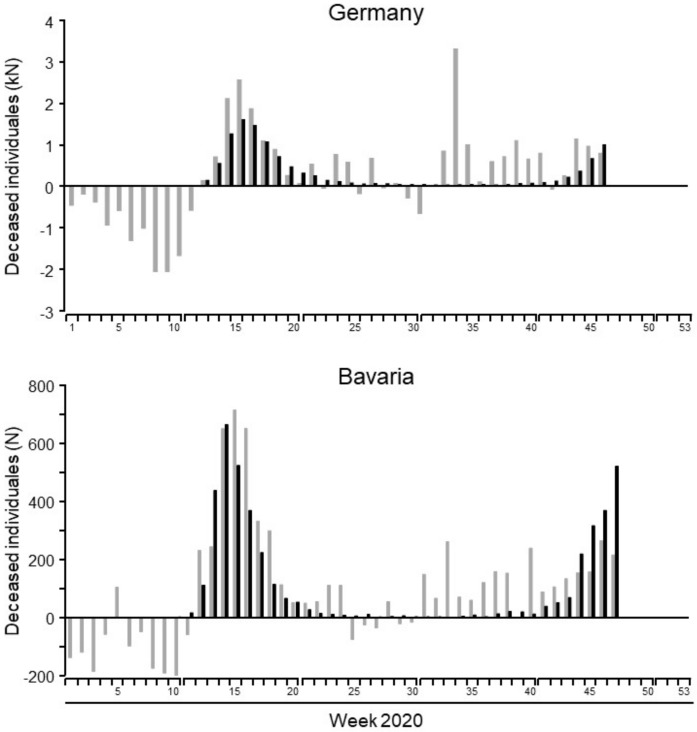
Excess mortality of 2020 in relation to the average mortality 2016–2019 (grey columns) and COVID-19-caused mortality (black columns) in Germany and Bavaria per week of 2020. Data Figure 95. masks became mandaTable 19. patients and surging infection rates (Figure 6).

**Figure 6 healthcare-09-00338-f006:**
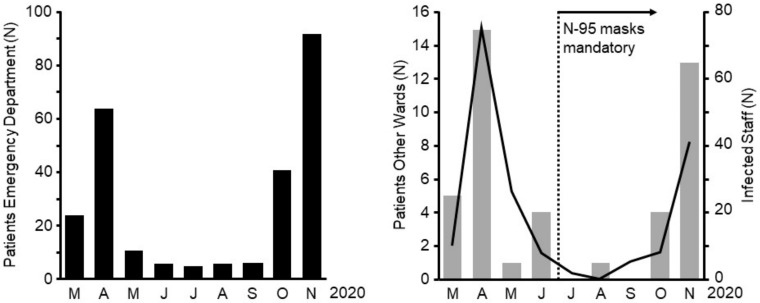
Number of SARS-CoV-2 infected patients treated in Ingolstadt hospital and number of SARS-CoV-2 infected Scheme 2. infection diagnosed at the emergency department (ED). Right: patients who were tested positive after admission to the hospital when being treated on other wards (grey bars, left *y*-axis) and number of staff members with confirmed SARS-CoV-2 infection (black line, right *x*-axis). Starting in late June, wearing N-95 (FFP-2) masks was introduced as mandatory during direct patient interactions for all staff members of Ingolstadt Hospital.

**Table 1 healthcare-09-00338-t001:** Estimated parameters of the 3-phase model for Germany, Bavaria, the city of Ingolstadt, and the Ingolstadt region. In all four cases, we see an increase of the shift parameter from the first to the second wave, and a decrease of the case-fatality rates. Comparing Ingolstadt and the neighboring region, it is striking that the case-fatality rates flip order, the Ingolstadt region having by far the highest case-fatality rate of all four considered entities in the fall of 2020.

Country	$\alpha_{1}$	$d_{1}$	Start Uncoupling	End Uncoupling	$\alpha_{2}$	$d_{2}$
Germany	0.045	14	25 June 2020	13 October 2020	0.015	18
Bavaria	0.048	14	10 July 2020	14 October 2020	0.014	16
Ingolstadt	0.077	11	17 May 2020	21 October 2020	0.011	20
Ingolstadt Region	0.06	9	13 May 2020	29 October 2020	0.031	17

**α_1_** = case-fatality rate in spring 2020 (first wave). **α_2_** = case-fatality rate in the fall of 2020 (second wave). ***d*_1_** = shift in days between reported infection and death in spring 2020 (first wave). ***d*_2_** = shift in days between reported infection and death in spring 2020 (second wave).

## Data Availability

Not applicable.

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
