# Peer review of "Incidence and Death Rates from COVID-19 Are Not Always Coupled: An Analysis of Temporal Data on Local, Federal, and National Levels"

_healthcare, 2021, doi:10.3390/healthcare9030338_

Round 1

Reviewer 1 Report

Thank you for the opportunity to read and review your work. I commend the authors for describing and providing evidence in support of a model for predicting COVID-19 case fatality based on locally collected incidence data. The significance of this includes providing an evidence-based tool to aide in tailoring local responses to mitigate COVID-19 spread and outbreaks to local epidemiological data.

I have some minor revision requests:

Abstract

  • p1, 4th line of abstract – there’s a small typo “…oft he…”, I believe should be “…of the…”

Introduction

  • p1, 3rd paragraph – could the authors cite a source in support of the sentence, “Worldwide social restriction measures as well as closure of restaurants and shops to minimize infections are currently regulated based on SARS-CoV-2 infection numbers during a 7-day period per 100,000 people.”?

Discussion

p9, 10th-12th lines of the Discussion – the authors write, “The reason for less hospital admissions and SARS-CoV-2 related fatalities despite rising infection rates in the summer remains unknown at this point.” The authors raise herd immunity as one possible explanation for this discontinuity.

  • The manuscript would be improved here if the authors included an acknowledgement that this phenomenon was observed internationally and was not limited to Germany or Bavaria. For example, in the U.S., beginning in the summer of 2020, healthcare workers also saw unprecedented increases in COVID-19 diagnoses and hospitalizations, but there wasn’t a congruent rise in mortality rates even as case counts set records. In fact, the COVID-19 mortality rate in the U.S. since the start of the pandemic fell.

  • In addition, as the authors mention in the discussion, that decline had no single, clear explanation. The manuscript would be improved by describing some other potential contributing factors, beyond herd immunity, including: a higher proportion of cases among the young, increased knowledge of how to treat COVID patients effectively, better therapies, and less overcrowding in hospitals.

  • The authors should add to the limitations that generalized implementation of the model by others would be dependent on accurate collection and reporting of local data.

Author Response

Dear reviewer, 

we thank you for your valuable comments and we hope that you will find our revisions satisfactory.

Reviewer 2 Report

Thank you for the authors' effort. Please consider review comments.

Author Response

Dear reviewer,

we thank you for your valuable comments and we hope that you find our revisions satisfactory.

Please see attacment

Reviewer 3 Report

The authors present a longitudinal analysis of incidence and death rates from COVID-19 at local, federal and national levels. Their proposal of using CFR as a metric to consider in order to make decisions regarding the pandemic seems interesting. However, the results are presented in an unclear way and the manuscript is difficult to follow. The discussion session is very weak and seems to be unconnected with the results. The main weakness is the lack of analysis of the patients pathways and treatments.  The clinical and organizational innovation in the studied period has been huge and has affected greatly the fatality ratios. This factor is overlooked. 

The reflections on herd immunity, genetic variants and reinfections are not well structured and somewhat anecdotical. 

The conclusions do not reflect the study findings and include statements that cannot be supported by the research design, such as "I-95 masks seem to grant superior protection...".

To sum up, the authors should focus on their analysis and do not try to go beyond it and, consequently, the sections Discussion and Conclusion should be rewritten.

Author Response

Dear reviewer,

we thank you for your valuable comments and we hope you find our revisions satisfactory.

Round 2

Reviewer 3 Report

The authors have provided adequate answers and improved the manuscript that is publishable in its current version.

This manuscript is a resubmission of an earlier submission. The following is a list of the peer review reports and author responses from that submission.

Round 1

Reviewer 1 Report

The authors forecast the death rates using the linear regression and analyzing the time-series COVID-19 incidence data. The manuscript is overall well written. However, I have major concerns that would need to be addressed and these concerns can be summarized in two points:

1. I am not convinced that predicting the death rates assuming linear relationship between incidence and death rates. The virus appears to result in mild symptoms or even no symptoms for the majority of young people, while the high case fatality rate has been reported in the elderly globally. For example, even though the number of newly infected cases is same, cluster in the nursing home and elementary school will show totally different number of COVID-19 related deaths. Therefore, I would incline that the simple prediction with the linear relationship assumption may not be plausible, without considering the age dependent IFR.

2. The naïve CFR (i.e., the proportion of the cumulative number of deaths out of the cumulative number of cases at a point in time) tends to be an underestimate due to the right-censoring issue. In addition, especially in the early stage of epidemic, right-censoring issue with respect to the time delay from illness onset to death may highly lead to underestimation of the CFR due to the exponential growth of incidence. Thus, I would incline that the authors should consider the right-censoring issue, rather using the naïve CFR calculation method.

Reviewer 2 Report

The subject is very interesting but it is important to improve some aspects:
1) the introduction is very short and I don't know what is clear, why is it important to do the study? what is the OBJETIVE? What is the contribution of the article to the literature? Would it help reduce the number of deaths? Does it help to better plan health policies ?; 2) the authors talk about a model to predict, but do not present it in the document, what are the parameters? What is the reliability of the model ?; 3) the discussion is not very clear. How to use the analysis in a region or city of Germany to extend it to Europe ?; 4) why not use a panel data analysis? one model for Eastern Europe and another for Western Europe; 5) I think they would give more empirical support to all the authors' arguments, which shows a good knowledge of the subject; 6) the explanations of the divergence by the authors seem reasonable but there is a lack of clarity throughout the document; from the title to the conclusions; 7) the authors should express the limitations of the article and future lines of research on this topic.

Reviewer 3 Report

In this manuscript, Stefan Borgmann et al developed a time-series model by the data from the European health authorities predicting death rates based on SARS-CoV-2 infection rates. The temporal relation of COVID-19 incidence and COVID-19 caused deaths were analyzed by a time series model. They analyzed data for 36 European countries to assess the epidemiology of in Europe and the current risk for local populations to die from COVID-19. They found that SARS-CoV-2 has spread from Western to Eastern Europe in the past months of July and August. Western European countries' case fatality rates decrease with rising infection numbers during the course of the pandemic. They deem that local health care systems seem to be safe in terms of severe disease and COVID-19 related ICU admissions once the locally reported incidence of death from COVID-19 falls under the predicted death curve. I think the article has clear logic and smooth writing and should be published except that there is an extra space between 1.2 and % in the line39.

Reviewer 4 Report

This is a timely contribution on the correlation of the infection rates and fatal cases, the latter decreasing during the recent second wave. Whereas the data as such warrant publication due to the consequences for the health care systems, the so called predictive model is debatable. It apparently fails, if the trend changes due to unexpected events (in France, when the death rates increased it failed).

Since the data as such indicate lower fatalities per infected patients the conclusion is convincing that there will be no increasing consequences for the health care system.

Thus, it is recommended to publish the data as such without the model. This would eliminate unnecessary repetitions.

Reviewer 5 Report

The article is clear, precise and concise.

The study addresses an extremely important issue related to the understanding of a health phenomenon related to the SARS-CoV-2 pandemic, which we are currently experiencing worldwide and which we still have much to learn. 

The authors seek to make a comparative analysis of the distribution of the disease and mortality in different European countries, in 2 different moments of time: an initial one at the beginning of the pandemic, April 2020 and a second moment, August of the same year. 

This analysis is based on official data on incidence and mortality, despite having some limitations, they are duly pointed out and recognized by the authors. 

The authors also present us with an attempt to estimate fatality rate from incidence data, which seems to be reliable, this estimate can be very useful in defining the health policy of each country. 

The data presented here are still a very important contribution to the understanding of the behavior of the pandemic in the different regions of Europe, in a way they also allow to evaluate the response that is given by the different European countries to the pandemic situation. 

The explanations raised by the authors to justify the differences observed in the distribution of the disease and mortality, in the different regions and in the two moments when the evaluation are very interesting, although in my opinion, are merely speculative, I think that there is still a lot of data to clarify . 

A less positive aspect of this article relates to the methods section, this section should be better explained, namely with regard to data sources and their collection and methods of data analysis.

In summary, the study is an attempt to understand a phenomenon about which we still know very little, there is much more to clarify than what we know at the moment.